# Mothers' utilization and associated factors of preconception care in Africa, a systematic review and meta-analysis

**Tiwabwork Tekalign**[1]*, **Tesfanesh Lemma**[2], **Mulualem Silesh**[2], **Eyasu Alem Lake**[1], **Mistire Teshome**[1], **Tesfaye Yitna**[1], **Nefsu Awoke**[1]

**1** School of Nursing, College of Health Science and Medicine, Wolaita Sodo University, Wolaita Sodo, Ethiopia, **2** Department of Midwifery, College of Health Science and Medicine, Debre Berhan University, Debre Berhan, Ethiopia

* tiwabworkt@gmail.com

## Abstract

**Data Availability Statement:** The data analyzed during the current systematic review and meta-analysis is available as Supporting information files.

### Background

As the studies show, in every minute in the world, 380 women become pregnant and 190 face unplanned or unwanted pregnancies; 110 experience pregnancy-related complications, and one woman dies from a pregnancy-related cause. Preconception care is one of the proven strategies for the reduction in mortality and decreases the risk of adverse health effects for the woman, fetus, and neonate by optimizing maternal health services and improves woman's health. Therefore, this study aimed to estimate the pooled prevalence of utilization of preconception of care and associated factors in Africa.

### Methods

Systematic search of published studies done on PubMed, EMBASE, MEDLINE, Cochrane, Scopus, Web of Science CINAHL, and manually on Google Scholar. This meta-analysis follows the Preferred Reporting Items for Systematic Reviews and Meta-Analyses (PRISMA) guidelines. The quality of studies was assessed by the modified Newcastle-Ottawa Scale (NOS). Meta-analysis was carried out using a random-effects method using the STATA™ Version 14 software.

### Result

From 249,301 obtained studies, 28 studies from 3 African regions involving 13067 women included in this Meta-analysis. The overall pooled prevalence of utilization of preconception care among pregnant women in Africa was found to be 18.72% (95% CI: 14.44, 23.00). Knowledge of preconception care (P = <0.001), preexisting medical condition (P = 0.045), and pregnancy intention (P = 0.016) were significantly associated with the utilization of preconception care.

**Funding:** The authors received no specific funding for this work.

**Competing interests:** All authors declared they had no competing interests.

**Abbreviations:** CI, Confidence Interval; NCD, Non Communicable Disease; NOS, Newcastle Ottawa Scale; OR, Odds Ratio; PCC, Preconception care; PRISMA, Preferred Reporting Items for Systematic Reviews and Meta-Analyses; WHO, World Health Organization.

## Conclusion

The results of this meta-analysis indicated, as one of best approaches to improve birth outcomes, the utilization of preconception care is significantly low among mothers in Africa. Therefore, health care organizations should work on strategies to improve preconception care utilization.

## Introduction

According to the World Health Organization (WHO), preconception care (PCC) is, the provision of biomedical, behavioral, and social health interventions to women and couples before conception occurs, to improve their health status, and mitigating behaviors, individual and environmental factors that could contribute to poor maternal and child health outcomes [1, 2]. This is done through risk identification, health education, and promotion, and initiation of evidence-based interventions in the period before conception. The use of PCC in high- and low-income countries aims to improve maternal pregnancy and neonatal outcomes both in the short and long term [3]. PCC also includes the detection and optimal control of specific medical conditions to optimize pregnancy-related outcomes for the woman and her offspring as well as implemented to prevent pregnancies that are unplanned, too early, or too close [4, 5].

Now a day promoting and enhancing women's health before pregnancy has a favorable outcome and highly reduces pregnancy and childbirth related complications [6], and also Preconception care can make a useful contribution to reducing maternal and childhood mortality and morbidity, and to improving maternal and child health in both high- and low-income countries [1]. In 2015, 303, 000 women in the world died from pregnancy and childbirth-related problems [7]. In Ethiopia, the pregnancy-related mortality ratio was 412 per 100,000 live births and the lifetime risk of pregnancy-related death is 21 in 1000 women [8]. Most of these complications develop during pregnancy, exist before, and worsened during pregnancy, especially if not managed as part of the PCC [2]. There is growing evidence that preconception care may have an important role in preventing short and long-term adverse health consequences for women and their offspring [9].

Besides PCC is very crucial for women with underlying chronic diseases, according to global statistics, non-communicable diseases (NCDs) are the cause of more than 53% of diseases. Moreover, it is predicted that NCDs will be the cause of 73% of deaths worldwide and 80% of deaths in developing countries [10, 11]. Researchers reported in their studies that 17.5% and 32% of pregnant mothers who were referred to healthcare centers had received pre-pregnancy care. Similarly, the findings of the studies conducted by Asresu and Betra 18.2% and 29.7% of people seek pre-pregnancy care programs [12–15]. In another study conducted by Frey and Files in Mayo clinic [16] found that only 39% of the women received PCC from their primary care physicians compared to 98.6% who believed in its importance.

There has been an increasing burden of maternal, newborn, and child mortality globally. Worldwide, 400/100000 women of childbearing age die every year due to complications of pregnancy and childbirth and 7 million infants die each year between birth to 12 months [17]. According to statistics, every minute in the world, 380 women become pregnant and 190 faces unplanned or unwanted pregnancies; 110 experiences a pregnancy-related complication; 40 have an unsafe abortion; and one woman dies from a pregnancy-related cause. Implementation of evidence-based preconception interventions improves infant and maternal pregnancy

outcomes [18]. This review will contribute to the integration of preconception care with other existing health programs, assignment of the task of pre-pregnancy health promotion to the healthcare workers, improvement or promotion of preconception services, engagement of the media, usage of healthcare information technology, maximizing demand for and uptake of preconception interventions, especially by adolescents.

## Objectives of the review

- To determine the prevalence of utilization of Preconception care in Africa

- To identify the associated factors of utilization Preconception care in Africa

## Methods and materials

### Study design and search strategy

We registered the protocol in PROSPERO (ID: CRD42020209551). This systematic review and meta-analysis was conducted under the guidelines of the Preferred Reporting Items for Systematic Reviews and Meta-analyses (PRISMA) statement [19, 20].

A three-step search strategy was utilized in this review. An initial limited search of PubMed was undertaken followed by the analysis of the text words contained in the title and abstract, and of the index terms used to describe the article. A second search was done by using all identified keywords and index terms across all included databases. Thirdly, the reference list of all identified reports and articles was searched for additional studies. Studies published in English language up to May 2021 were taken from EMBASE, MEDLINE, Cochrane, Scopus, Web of Science, CINAHL, and manually on Google Scholar. The search for unpublished studies included Google and institutional repositories. The search was performed using key terms such as preconception care, PCC, Pre-pregnancy care, prenatal care, folic acid, multi-vitamin, foliate supplement, folic acid intake, Iron–folic acid, IFA, Mother, reproductive age group, pregnant women, utilization, and uptake.

### Study selection and eligibility criteria

- Participants in the studies should be mothers.

- Both published and unpublished studies conducted in Africa were included.

- Studies that reported the prevalence of utilization of preconception care among mothers regardless of study design

### Study extraction and quality appraisal

The data were extracted by three independent authors (TT, MT, and T.L) using a data extraction format prepared in a Microsoft Excel 2010 spreadsheet. The extracted data were: the first author's name, publication year, country, design, sample size, sampling method, utilization of preconception care, and associated factors with their odds ratio. The quality of each study was assessed using the modified Newcastle-Ottawa Scale (NOS) for cross-sectional studies [21, 22] Studies were included with a score of 5 and more on the NOS [23]. The quality of each study was evaluated independently by four authors (TT, NA, MT, and T.L) and ay disagreements were resolved by discussion and consensus.

## Publication bias and heterogeneity

To assess the existence of publication bias, funnel plots were used and Egger's test was computed. A p-value< 0.05 was used to declare the statistical significance of publication bias. I2 test statistics were used to check the heterogeneity of studies. $I^2$ test statistics of < 50, 50–75% and > 75% was declared as low, moderate and high heterogeneity respectively [24].

## Outcome measure

The primary outcome of this review was the prevalence of utilization of preconception care. The second outcome of this review was the associated factors of preconception care utilization. The only factor identified as a significant factor in the two and above studies was included in this review and meta-analysis.

## Data synthesis and analysis

STATA™ Version 14 software was used to conduct the analysis. The heterogeneity test was conducted by using I-squared ($I^2$) statistics. The pooled prevalence of utilization of preconception care was carried out using a random-effects (DerSimonian and Laird) method. To minimize the potential random variations between studies; the sources of heterogeneity were analyzed using subgroup analysis, and meta-regression. A sensitivity analysis was also conducted.

# Results

## Study selection

Initially, a total of 249,301 studies were retrieved from the databases and manual searching. From this, 17195 duplicates were found and removed. The remaining 232,106 articles were screened by their title and abstract and 231378 irrelevant studies were removed. 728 full-text articles were assessed for eligibility and 700 of them were excluded due to not reporting the outcome of interest, which doesn't report the computed value of the outcome of interest. Finally, a total of 28 studies was fulfilled the inclusion criteria and enrolled in the study (Fig 1).

## Study characteristics

The 28 studies [25–51] included 13067 participants. All of the included studies were cross-sectional studies and the sample size ranged from 50 [31] to 1331 [43]. Most studies were conducted in Ethiopia. Among the included studies, utilization of preconception care among mothers were ranged from 2.5 [40] to 86.8 [31] (Table 1).

## Utilization of preconception care among mothers

By including the twenty-eight published research articles we had estimated the pooled prevalence utilization of preconception care among mothers in Africa. Accordingly, the overall estimated pooled prevalence of utilization of preconception care among mothers with a random-effects model was 18.72% (95% CI: 14.44, 23.00) with a heterogeneity index ($I^2$) of 98.7% (p = 0.000) (Fig 2).

## Subgroup analysis

Subgroup analyses revealed a marked variation across regions. Based on the subgroup analysis result, the highest (24.81%; 95% CI: 14.80, 34.82), $I^2$ = 99.3%) seen in western region and the lowest (15.90%; 95% CI: 11.54, 20.26), $I^2$ = 97.6%) seen in eastern region (Fig 3).

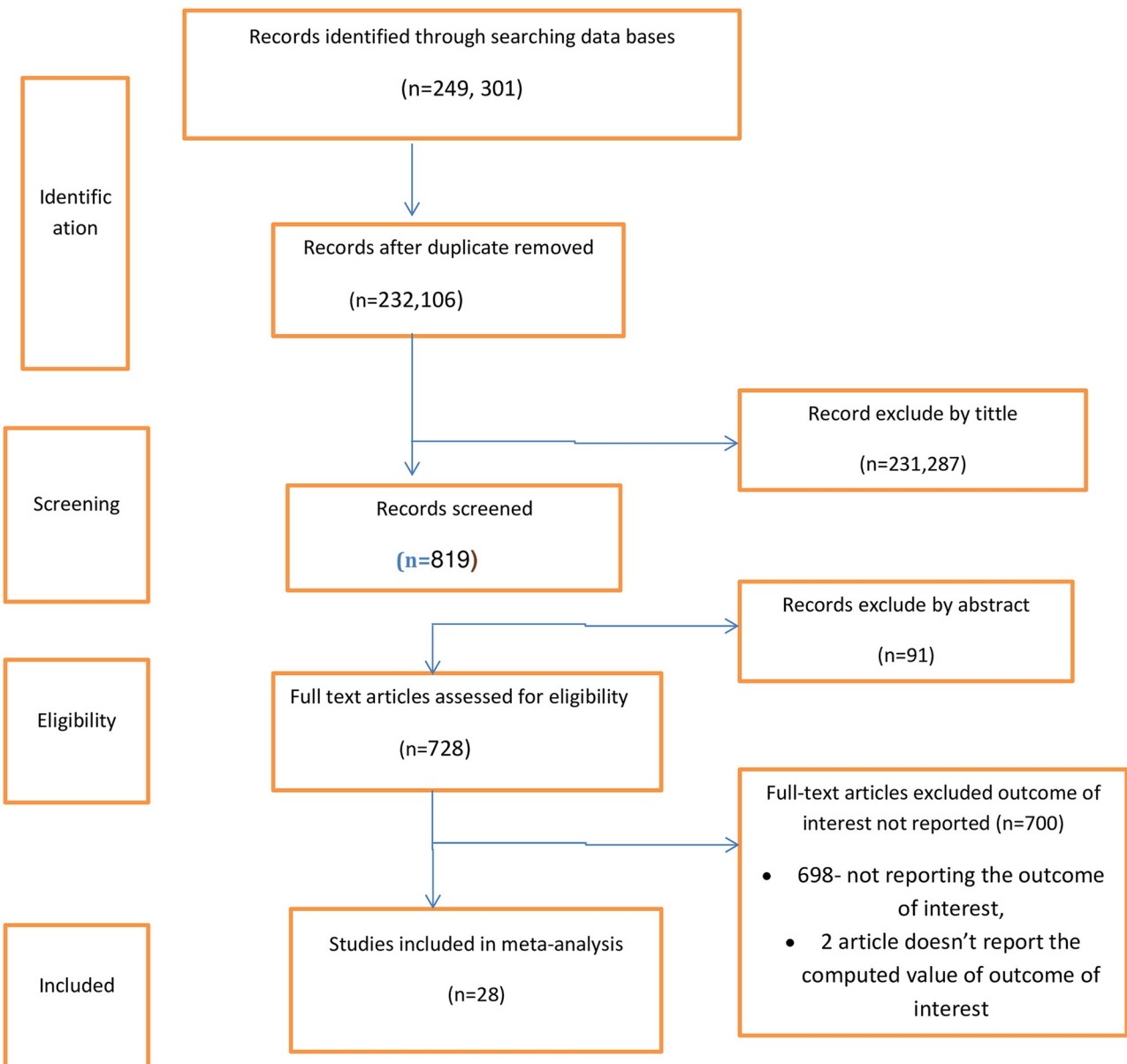

**Fig 1. PRISMA flow diagram of study selection.**

## Heterogeneity and publication bias

Meta-regression was conducted to identify the source of heterogeneity using sample size as a covariate (Table 2). It was indicated that there is no effect of sample size on heterogeneity between studies. The presence of publication bias was checked using the Egger's test, and graphical by Funnel plot, the result egger's test was found significant (p<0.000), as a result to estimating the number of missing studies that might exist in a meta-analysis we conducted Duval and Tweedie's trim and fill analysis, but is not significant. Also, visual inspection of the funnel plot indicated asymmetrical distribution showing publication bias (Fig 4).

**Table 1. Characteristics of the included studies in the systematic review and meta-analysis.**

| Authors Name | Publication Year | Study setting | Country | Study design | sample | prevalence%(95% CI) |
|---|---|---|---|---|---|---|
| Goshu, Y. A., et al | 2018 | Adet town | Ethiopia | Cross-sectional | 229 | 9.6(5.78–13.41) |
| Asresu, T. T., et al | 2019 | Mekelle City | Ethiopia | Cross-sectional | 561 | 18.2(15.0–21.39) |
| Demisse, T. L., et al | 2019 | Debre Birhan Town | Ethiopia | Cross-sectional | 410 | 13.4(10.10–16.69) |
| Okemo, J., | 2020 | Aga Khan University Hospital & Maragua Level Four Hospital | Kenya | Cross-sectional | 194 | 25.8(19.64–31.95) |
| Fekene, D. B., et al | 2020 | west shoa | Ethiopia | Cross-sectional | 669 | 14.5(11.83–17.16) |
| Metasebia Getachew | Unpublished | Debre Berhan | Ethiopia | Cross-sectional | 413 | 16.5(12.92–20.07) |
| Olowokere, A.E et al | 2015 | Osun State | Nigeria | Cross-sectional | 375 | 34.1(29.30–38.89) |
| Akinajo, O. R et al | 2020 | Lagos | Nigeria | Cross-sectional | 50 | 86.8(77.41–96.18) |
| Gezahegn, A. (2016) | Unpublished | west Shoa Zon | Ethiopia | Cross-sectional | 634 | 38.2(34.41–41.98) |
| Napoleon N. Ekem, et al | 2018 | teaching hospital Abakaliki | Nigeria | Cross-sectional | 453 | 10.3(7.50–13.09) |
| Adeyemo, A. A., & Bello, O. O. | 2021 | University College Hospital, Ibadan | Nigeria | Cross-sectional | 414 | 18.8(15.03–22.56) |
| Taddese F. | Unpublished | St. Paul's Millennium Medical College | Ethiopia | Cross-sectional | 280 | 18.1(13.59–22.6) |
| Setegn M. | 2021 | Mizan Aman | Ethiopia | Cross-sectional | 605 | 16.2(13.26–19.13) |
| Teshome F, | 2021 | Manna District | Ethiopia | Cross-sectional | 623 | 6.3(4.39–8.20) |
| Lawal TA, Adeleye AO. | 2014 | Ibadan | Nigeria | Cross-sectional | 602 | 2.5(1.25–3.74) |
| Alsammani MA, et al | 2017 | Sudan | Sudan | Cross-sectional | 1000 | 3.2(2.10–4.29) |
| Boakye-Yiadom AK, et al | 2020 | Tamale west hospital | Ghana | Cross-sectional | 200 | 15(10.05–19.94) |
| Ahmed K, et al | 2015 | Sudan | Sudan | Cross-sectional | 100 | 40(30.39–49.6) |
| Ezegwui HU, | 2008 | Nigeria | Nigeria | Cross-sectional | 1331 | 47.7(45.01–50.38) |
| Al Darzi W, et al | 2014 | Ain Shams University Hospital | Egypt | Cross-sectional | 660 | 8.8(6.63–10.96) |
| Dessie MA, et al | 2017 | Adama hospital medical college | Ethiopia | Cross-sectional | 417 | 3.5(1.73–5.26) |
| Okon UA, et al | 2020 | Benue State | Nigeria | Cross-sectional | 586 | 27.6(23.98–31.21) |
| Abdulmalek LJ. | 2017 | Benghazi | Libya | Cross-sectional | 131 | 6(1.93–10.06) |
| Senoga I. | Unpublished | KCCA health centers, Kampala. | Uganda | Cross-sectional | 423 | 16.5(12.96–20.03) |
| Adebo OO, et al | 2017 | Nigeria | Nigeria | Cross-sectional | 300 | 3(1.06–4.93) |
| Anzaku AS. | 2013 | Jos | Nigeria | Cross-sectional | 543 | 7.4(5.19–9.60) |

*(Continued)*

**Table 1.** (Continued)

| Authors Name | Publication Year | Study setting | Country | Study design | sample | prevalence%(95% CI) |
|---|---|---|---|---|---|---|
| C A. ENUKU, & FO. Adeyemo | 2019 | delta state | Nigeria | Cross-sectional | 273 | 24.2(19.11–29.28) |
| Habte A, et al | 2020 | Southern state | Ethiopia | Cross-sectional | 591 | 6.4(4.42–8.37) |

## Sensitivity analysis

Sensitivity analysis was done by removing studies step by step to evaluate the effect of a single study on the overall effect estimate. The result indicated removing a single study did not have a significant influence on pooled prevalence (Fig 5).

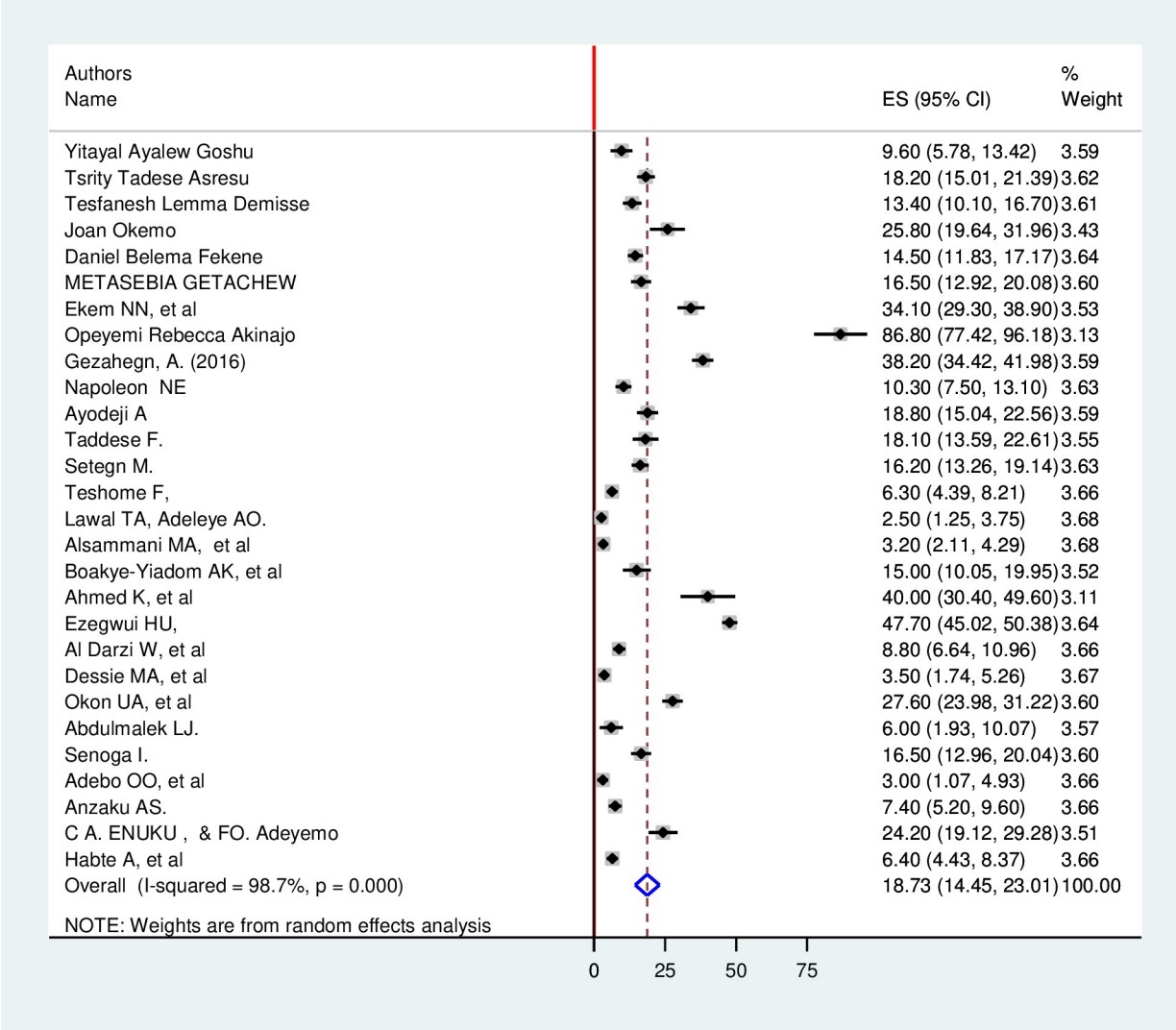

**Fig 2. Forest plot showing pooled prevalence of utilization of preconception care among women in Africa.**

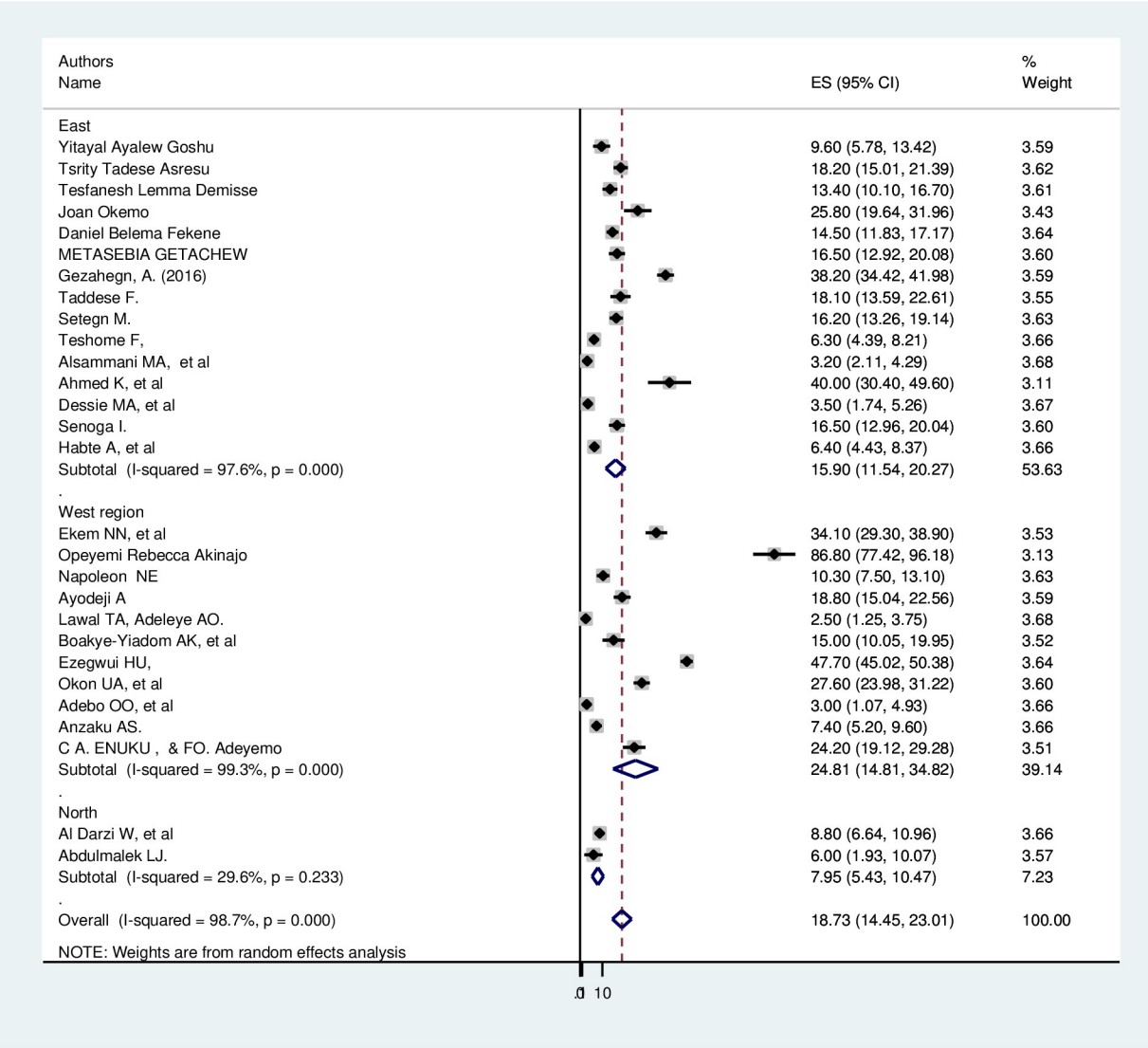

**Fig 3. Subgroup analysis of utilization of preconception care among mother by country in Africa.**

## Factors associated with utilization of preconception care

Six variables were extracted to identify factors affecting the utilization of preconception care among women. Of these, three variables (knowledge, Pre-existing medical condition, and pregnancy intention) were found to be significantly associated with utilization of preconception care (Table 3).

Mothers having poor knowledge of preconception care were 39% less likely to utilize the care than those having good knowledge (OR: 0.61(95% CI 0.51–0.74), p = 0.000, $I^2$: 97.5%, the heterogeneity test (p< 0.001). Those mothers who had pre-existing medical condition were

**Table 2. Meta-regression analysis of factors affecting between-study heterogeneity.**

| Heterogeneity source | Coefficients | Std. Err. | P-value |
|---|---|---|---|
| Sample size | -0.0031257 | .0052047 | 0.553 |

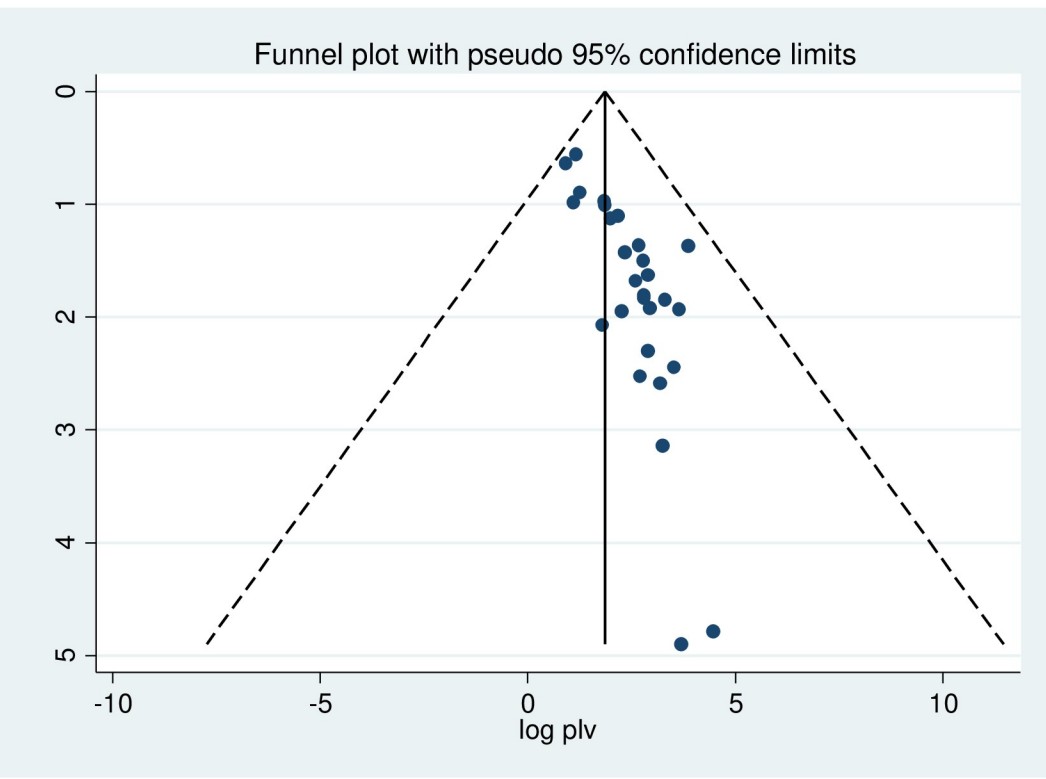

**Fig 4. Funnel plot to test the publication bias in 28 studies with 95% confidence limits.**

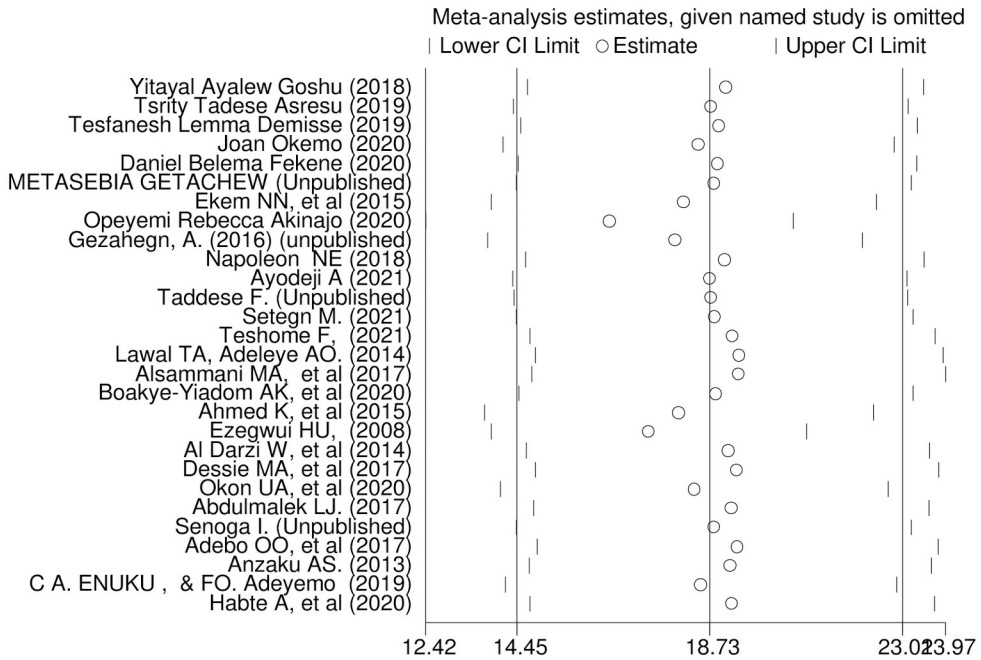

**Fig 5. Sensitivity analysis of pooled prevalence for each study being removed one at a time.**

**Table 3. Factors associated with utilization of preconception care among women in Africa.**

| Determinants | Comparison | Number of studies | Sample size | OR(95%CI) | P- value | $I^2$ (%) | Heterogeneity test (p value) |
|---|---|---|---|---|---|---|---|
| Knowledge | Poor Vs Good Knowledge | 7 | 3581 | 0.61(0.51–0.74) | < 0.001 | 97.5 | < 0.001 |
| Educational status | No formal education Vs Primary school and above | 7 | 3475 | 0.68(0.44–1.05) | 0.084 | 80.0 | 0.007 |
| Marital status | Single Vs others | 3 | 1103 | 0.68(0.44–1.05) | 0.084 | 80.0 | 0.007 |
| Pre-existing medical condition | Yes vs No | 4 | 1615 | 0.71(0.51–0.99) | 0.042 | 96.7 | < 0.001 |
| Adverse birth outcome | Yes vs No | 2 | 845 | 1.15(0.81–1.64) | 0.416 | 98.7 | < 0.001 |
| Pregnancy intension | Yes vs No | 3 | 1247 | 2.47(1.74–3.52) | 0.000 | 75.9 | 0.016 |

29% less likely to utilize preconception care than their counterparts (OR: 0.71(95% CI 0.51–0.99), P = 0.045, $I^2$: 96.7%, the heterogeneity test (p< 0.001).

Mothers who had pregnancy intention were 2.5 times more likely to utilize preconception care than those who hadn't have an intention (OR: 2.47(95% CI 1.74–3.52), P = 0.000, $I^2$: 75.9%, the heterogeneity test (p = 0.016).

## Discussion

Many medical conditions, personal behaviors, psychosocial risks, and environmental exposures associated with negative pregnancy outcomes can be identified and modified before conception through clinical interventions. For certain conditions, opportunities for preventive interventions occur only before conception, this is by preconception care. PCC is one way believed to improve pregnancy outcomes and is considered important by health care workers and the general population [52, 53].

A systematic review of tools to assess the quality of observational studies examining incidence or prevalence concluded that there is no consensus exists as to which individual criteria should be assessed to establish methodological quality [54].

The Cochrane Collaboration advises assessing the risk of bias on a subjective basis using domain-based evaluation [55], so we used the Newcastle-Ottawa Quality Assessment Scale (adapted for cross-sectional studies) [22] and authors independently reviewed with minimal disagreement between reviewers.

According to this systematic review and meta-analysis, the estimated pooled prevalence of utilization of preconception care among mothers was 18.72% (95% CI: 14.44, 23.00). It was believed that preconception care helps to fill the gap in the existing continuum of maternal and child healthcare [56]. Different randomized controlled trials also showed that one of PCC component folate supplementation (alone, or in combination with other vitamins and minerals) reduces the prevalence of neural tube defect [57–59].

Every woman of reproductive age who is capable of becoming pregnant is a candidate for preconception care, regardless of whether she is planning to conceive [60]. But PCC implementation is in the infant stage in low and middle-income countries [61].

Based on the subgroup analysis result, the highest (24.81%; 95% CI: 14.80, 34.82), I2 = 99.3%) seen in western region and the lowest (15.90%; 95% CI: 11.54, 20.26), I2 = 97.6%) seen in eastern region. The difference might be because of the difference in sample size, socio-economic status of the countries, and the number of included studies in this meta-analysis.

Among the extracted factors preconception care knowledge, pre-existing medical condition, and pregnancy intention were found to be significantly associated with utilization of preconception care.

Seven studies revealed that having adequate knowledge about PCC was strongly associated with the utilization of PCC. Mothers having poor knowledge of preconception care were 39% less likely to utilize the care than those having good knowledge (OR: 0.61(95% CI 0.51–0.74); this is consistent with a systematic review conducted in Ethiopia [62]. knowing enhances the utilization of any health-related service.

Having pre-existing medical conditions has an impact on the utilization of PCC. In this study, those mothers who had pre-existing medical conditions were 29% less likely to utilize preconception care than their counterparts (OR: 0.71(95% CI 0.51–0.99). this might be explained as those having pre-existing medical conditions entirely worry about their medical condition than using preconception care service. Also, a study conducted in Nigeria showed that none of the participants with pre-existing medical conditions had awareness of PCC [63].

Pregnancy intention is one of the means which facilitates using of preconception care service. According to this study, mothers who had pregnancy intention were 2.5 times more likely to utilize preconception care than those who hadn't have the intention (OR: 2.47(95% CI 1.74–3.52). If women had the intention to have a healthy baby, the probability of using PCC will increase.

## Conclusion

The results of this meta-analysis indicated as one of best approaches to improve birth outcomes, the utilization of preconception care is significantly low among mothers in Africa. Therefore, health care organizations should work on strategies to improve preconception care utilization.

## Limitation of the study

This systematic review and meta-analysis presented the prevalence of preconception care utilization in Africa; it might have faced the following limitations. First, the lack of studies from southern and middle Africa may affect the generalizability of the finding to Africa. Secondly, due to the presence of significant heterogeneity and presence of publication bias, the result should be interpreted cautiously. Finally, we have faced difficulties in comparing our findings due to the lack of regional and worldwide systematic reviews and meta-analysis.

## Supporting information

**S1 Checklist. PRISMA 2009 checklist.**
(DOC)

**S1 Data. Raw data.**
(XLSX)

## Acknowledgments

We would like to thank all authors of studies included in this systematic review and meta-analysis.

## Author Contributions

**Conceptualization:** Tiwabwork Tekalign.

**Data curation:** Tiwabwork Tekalign, Mistire Teshome.

**Formal analysis:** Tiwabwork Tekalign, Eyasu Alem Lake, Mistire Teshome, Tesfaye Yitna, Nefsu Awoke.

**Investigation:** Tiwabwork Tekalign, Tesfanesh Lemma.

**Methodology:** Tiwabwork Tekalign, Tesfanesh Lemma, Mulualem Silesh, Eyasu Alem Lake, Mistire Teshome, Tesfaye Yitna, Nefsu Awoke.

**Software:** Tiwabwork Tekalign, Tesfanesh Lemma, Mulualem Silesh, Eyasu Alem Lake, Mistire Teshome, Tesfaye Yitna, Nefsu Awoke.

**Supervision:** Tiwabwork Tekalign, Tesfanesh Lemma, Mulualem Silesh, Eyasu Alem Lake, Mistire Teshome, Tesfaye Yitna, Nefsu Awoke.

**Validation:** Tiwabwork Tekalign, Tesfanesh Lemma, Mulualem Silesh, Eyasu Alem Lake, Mistire Teshome, Tesfaye Yitna, Nefsu Awoke.

**Visualization:** Tiwabwork Tekalign, Mulualem Silesh, Eyasu Alem Lake, Mistire Teshome.

**Writing – original draft:** Tiwabwork Tekalign.

**Writing – review & editing:** Tiwabwork Tekalign.

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
