## [Decision Letter · Decision Letter 0]

20 May 2021

PONE-D-21-05218

Mothers’ utilization and associated factors of preconception care in Africa, a systematic review and meta-analysis

PLOS ONE

Dear Dr. Tefesse,

Thank you for submitting your manuscript to PLOS ONE. After careful consideration, we feel that it has merit but does not fully meet PLOS ONE’s publication criteria as it currently stands. Therefore, we invite you to submit a revised version of the manuscript that addresses the points raised during the review process.

The academic editor has served as the second reviewer on this manuscript and agree with the assessment that major revisions are needed. 

We look forward to receiving your revised manuscript.

Kind regards,

Joseph Telfair, DrPH, MSW, MPH

Academic Editor

PLOS ONE

Journal Requirements:

2. Please confirm that you have included all items recommended in the PRISMA checklist including the full electronic search strategy used to identify studies with all search terms and limits for at least one database.

4.Thank you for stating the following financial disclosure:

 "not applicable"

6. We noticed you have some minor occurrence of overlapping text with the following previous publication, which needs to be addressed:

- https://journals.plos.org/plosone/article?id=10.1371/journal.pone.0241226

The text that needs to be addressed involves the Results section.

In your revision ensure you cite all your sources (including your own works), and quote or rephrase any duplicated text outside the methods section. Further consideration is dependent on these concerns being addressed.

Reviewers' comments:

Reviewer's Responses to Questions

**Comments to the Author**

1. Is the manuscript technically sound, and do the data support the conclusions?

Reviewer #1: No

2. Has the statistical analysis been performed appropriately and rigorously? 

Reviewer #1: No

3. Have the authors made all data underlying the findings in their manuscript fully available?

Reviewer #1: Yes

4. Is the manuscript presented in an intelligible fashion and written in standard English?

Reviewer #1: Yes

5. Review Comments to the Author

Reviewer #1: The authors conducted a systematic review and meta-analysis focusing on an important topic: the prevalence and factors associated with preconception care utilization in Africa. I commend the efforts of the authors to provide a sound evidence to improve the maternal and child health in the African continents. However, at the present state, I believe the manuscript must be further improved in order to be accepted for publication. Please find my comments, which would hopefully help to improve the manuscript as follows:

1) The search terms used are not specified in the manuscript and the term “Africa” is too vague. How do you define the countries included in Africa.

2) Another major concern is the narrow focus placed on the term preconception care. Preconception care covers many different services such as genetic screening, nutritional advice, violence prevention etc. and by just using the term “preconception care” in the search strategy, I am afraid this would provide a false picture of the actual situation in Africa. For example, the authors got 240551 hits from the database search and in the final analyses, the authors included only 11 studies and I noticed that these studies specifically included the term “preconception care” in their titles.

I suggest the authors to define “preconception care” carefully and not relying only on the titles as I believe many programs that would come under the umbrella of preconception care have already been conducted in many sub-Saharan African countries.

3) In the study selection, I do not understand by the authors only included “observational” studies. For example, an RCT could also report the prevalence of utilization of preconception care. By excluding other types of studies without proper justification would really lead the authors to arrive at a wrong conclusion.

4) How were the factors associated with utilization of preconception care is not clear. The authors wrote that factors reported in two or more studies were included but I believe this is not a convincing justification.

5) The conclusion states that “preconception care is low among mothers in Africa”. After reviewing the manuscript and the methods employed, I am not at all convinced with this conclusion as only three countries were included (Ethiopia, Nigeria, and Kenya). The results from these three countries might not represent the true picture of the African continent.

I would suggest the authors to consider the above points and probably reanalyze the results to arrive at a more trustworthy conclusion, which would hopefully be an important evidence to improve the lives of mothers and children in Africa.

6. PLOS authors have the option to publish the peer review history of their article (what does this mean?). If published, this will include your full peer review and any attached files.

Reviewer #1: No

---

## [Author Response · Author response to Decision Letter 0]

3 Jun 2021

Response to reviewers 

Review Comments to the Author

1. Please ensure that your manuscript meets PLOS ONE's style requirements, including those for file naming. The PLOS ONE style templates

Response – corrected

2. Please confirm that you have included all items recommended in the PRISMA checklist including the full electronic search strategy used to identify studies with all search terms and limits for at least one database

Response – we confirm that we followed recommended PRISMA checklis

3. In your Data Availability statement, you have not specified where the minimal data set underlying the results described in your manuscript can be found. PLOS defines a study's minimal data set as the underlying data used to reach the conclusions drawn in the manuscript and any additional data required to replicate the reported study findings in their entirety. All PLOS journals require that the minimal data set be made fully available.

Response – corrected 

4. Thank you for stating the following financial disclosure: "not applicable"

a. Please clarify the sources of funding (financial or material support) for your study. List the grants or organizations that supported your study, including funding received from your institution.

d. If you did not receive any funding for this study, please state: “The authors received no specific funding for this work.”

Response – corrected 

Response – corrected 

6. Please include captions for your Supporting Information files at the end of your manuscript, and update any in-text citations to match accordingly. Please see our Supporting Information guidelines for more information: http://journals.plos.org/plosone/s/supporting-information

Response – corrected 

7. We noticed you have some minor occurrence of overlapping text with the following previous publication, which needs to be addressed:

- https://journals.plos.org/plosone/article?id=10.1371/journal.pone.0241226

The text that needs to be addressed involves the Results section.

In your revision ensure you cite all your sources (including your own works), and quote or rephrase any duplicated text outside the methods section. Further consideration is dependent on these concerns being addressed.

Response – corrected 

Reviewer #1: The authors conducted a systematic review and meta-analysis focusing on an important topic: the prevalence and factors associated with preconception care utilization in Africa. I commend the efforts of the authors to provide a sound evidence to improve the maternal and child health in the African continents. However, at the present state, I believe the manuscript must be further improved in order to be accepted for publication. Please find my comments, which would hopefully help to improve the manuscript as follows:

8. The search terms used are not specified in the manuscript and the term “Africa” is too vague. How do you define the countries included in Africa

Response – yes you wright, we just removed the word Africa 

9. Another major concern is the narrow focus placed on the term preconception care. Preconception care covers many different services such as genetic screening, nutritional advice, violence prevention etc. and by just using the term “preconception care” in the search strategy, I am afraid this would provide a false picture of the actual situation in Africa. For example, the authors got 240551 hits from the database search and in the final analyses, the authors included only 11 studies and I noticed that these studies specifically included the term “preconception care” in their titles. I suggest the authors to define “preconception care” carefully and not relying only on the titles as I believe many programs that would come under the umbrella of preconception care have already been conducted in many sub-Saharan African countries.

Response – corrected 

10. In the study selection, I do not understand by the authors only included “observational” studies. For example, an RCT could also report the prevalence of utilization of preconception care. By excluding other types of studies without proper justification would really lead the authors to arrive at a wrong conclusion.

Response – yes of course , you are right, we corrected on the revised version

11. How were the factors associated with utilization of preconception care is not clear. The authors wrote that factors reported in two or more studies were included but I believe this is not a convincing justification.

Response – we said the above statement because we are trying to identify associated factors, similar to multivariate analysis which needs to have more than two variables, in our case one variable should be identified as significant factor at least for two different articles. Also in case meta-analysis heterogeneity score is important which can be observed for articles conducted in at least two similar region ,(similarly for associated factors)

12. The conclusion states that “preconception care is low among mothers in Africa”. After reviewing the manuscript and the methods employed, I am not at all convinced with this conclusion as only three countries were included (Ethiopia, Nigeria, and Kenya). The results from these three countries might not represent the true picture of the African continent.

Response- in this version, we added studies from other parts of Africa and mentioned as a limitation.

---

## [Decision Letter · Decision Letter 1]

16 Jun 2021

PONE-D-21-05218R1

Mothers’ utilization and associated factors of preconception care in Africa, a systematic review and meta-analysis

PLOS ONE

Dear Dr. Tefesse,

Thank you for submitting your manuscript to PLOS ONE. After careful consideration, we feel that it has merit but does not fully meet PLOS ONE’s publication criteria as it currently stands. Therefore, we invite you to submit a revised version of the manuscript that addresses the points raised during the review process.

ACADEMIC EDITOR: Before consideration for publication to be finally considered, the authors are asked to submit the manuscript to an editor to address grammatical errors, specifically English language usage. 

We look forward to receiving your revised manuscript.

Kind regards,

Joseph Telfair, DrPH, MSW, MPH

Academic Editor

PLOS ONE

Journal Requirements:

Additional Editor Comments (if provided):

Before consideration for publication to be finally considered, the authors are asked to submit the manuscript to an editor to address grammatical errors, specifically English language usage.

Reviewers' comments:

Reviewer's Responses to Questions

**Comments to the Author**

1. If the authors have adequately addressed your comments raised in a previous round of review and you feel that this manuscript is now acceptable for publication, you may indicate that here to bypass the “Comments to the Author” section, enter your conflict of interest statement in the “Confidential to Editor” section, and submit your "Accept" recommendation.

Reviewer #1: All comments have been addressed

2. Is the manuscript technically sound, and do the data support the conclusions?

Reviewer #1: Partly

3. Has the statistical analysis been performed appropriately and rigorously? 

Reviewer #1: Yes

4. Have the authors made all data underlying the findings in their manuscript fully available?

Reviewer #1: Yes

5. Is the manuscript presented in an intelligible fashion and written in standard English?

Reviewer #1: No

6. Review Comments to the Author

Reviewer #1: I think the revisions and responses are reasonable.

However, I find many typos throughout the manuscript.

I would strongly suggest the authors to submit this manuscript for extensive English editing.

7. PLOS authors have the option to publish the peer review history of their article (what does this mean?). If published, this will include your full peer review and any attached files.

Reviewer #1: No

---

## [Author Response · Author response to Decision Letter 1]

20 Jun 2021

Thank you for all constructive comments

 Response to reviewers 

Journal Requirements:

Response – revised 

1. Is the manuscript technically sound, and do the data support the conclusions?

Response – corrected 

2. Is the manuscript presented in an intelligible fashion and written in standard English?

Response – corrected

---

## [Decision Letter · Decision Letter 2]

29 Jun 2021

PONE-D-21-05218R2

Mothers’ utilization and associated factors of preconception care in Africa, a systematic review and meta-analysis

PLOS ONE

Dear Dr. Tefesse,

Thank you for submitting your manuscript to PLOS ONE. After careful consideration, we feel that it has merit but does not fully meet PLOS ONE’s publication criteria as it currently stands. Therefore, we invite you to submit a revised version of the manuscript that addresses the points raised during the review process.

ACADEMIC EDITOR: The Academic Editor served as the second reviewer on this manuscript.

Please attend to or clarify the requested correction in your results section in order to move forward for further publication consideration.

We look forward to receiving your revised manuscript.

Kind regards,

Joseph Telfair, DrPH, MSW, MPH

Academic Editor

PLOS ONE

Journal Requirements:

Additional Editor Comments (if provided):

Please attend to or clarify the requested correction in your results section in order to move forward for further publication consideration.

Reviewers' comments:

Reviewer's Responses to Questions

**Comments to the Author**

1. If the authors have adequately addressed your comments raised in a previous round of review and you feel that this manuscript is now acceptable for publication, you may indicate that here to bypass the “Comments to the Author” section, enter your conflict of interest statement in the “Confidential to Editor” section, and submit your "Accept" recommendation.

Reviewer #1: All comments have been addressed

2. Is the manuscript technically sound, and do the data support the conclusions?

Reviewer #1: Yes

3. Has the statistical analysis been performed appropriately and rigorously? 

Reviewer #1: Yes

4. Have the authors made all data underlying the findings in their manuscript fully available?

Reviewer #1: Yes

5. Is the manuscript presented in an intelligible fashion and written in standard English?

Reviewer #1: Yes

6. Review Comments to the Author

Reviewer #1: Thank you for your revisions and this will be my final comment.

The result presentation in the abstract is not correct.

Please check carefully before publication as this would affect the quality of your paper.

"Knowledge of preconception care (P=0. 61), preexisting medical condition (P=0.71), and

pregnancy intention (p =2.47) were significantly associated with the utilization of preconception

care"

If "P" refers to the p-value, they are not significant. I believe you are referring to the odds ratio.

7. PLOS authors have the option to publish the peer review history of their article (what does this mean?). If published, this will include your full peer review and any attached files.

Reviewer #1: No

---

## [Author Response · Author response to Decision Letter 2]

29 Jun 2021

Response to reviewers

Thank you for the comments 

Journal Requirements:

 Response - all references were corrected with the help of google scholar but reference number 9, was cited by the way authors suggestion, due to we are unable to got in google scholar , it is not retracted

About reference number, 22 , 57 – we used citation suggested in the journal but it differs in google scholar- so we corrected I accordingly

Review Comments to the Author

Reviewer #1: Thank you for your revisions and this will be my final comment.

The result presentation in the abstract is not correct.

Please check carefully before publication as this would affect the quality of your paper.

"Knowledge of preconception care (P=0. 61), preexisting medical condition (P=0.71), and

pregnancy intention (p =2.47) were significantly associated with the utilization of preconception

care"

If "P" refers to the p-value, they are not significant. I believe you are referring to the odds ratio.

Response- corrected

---

## [Decision Letter · Decision Letter 3]

7 Jul 2021

Mothers’ utilization and associated factors of preconception care in Africa, a systematic review and meta-analysis

PONE-D-21-05218R3

Dear Dr. Tefesse,

We’re pleased to inform you that your manuscript has been judged scientifically suitable for publication and will be formally accepted for publication once it meets all outstanding technical requirements.

Kind regards,

Joseph Telfair, DrPH, MSW, MPH

Academic Editor

PLOS ONE

Additional Editor Comments (optional):

The academic editor served as the second and final reviewer for this revision and agree its should be  accepted for publication.

Reviewers' comments:

Reviewer's Responses to Questions

**Comments to the Author**

1. If the authors have adequately addressed your comments raised in a previous round of review and you feel that this manuscript is now acceptable for publication, you may indicate that here to bypass the “Comments to the Author” section, enter your conflict of interest statement in the “Confidential to Editor” section, and submit your "Accept" recommendation.

Reviewer #1: All comments have been addressed

2. Is the manuscript technically sound, and do the data support the conclusions?

Reviewer #1: Yes

3. Has the statistical analysis been performed appropriately and rigorously? 

Reviewer #1: Yes

4. Have the authors made all data underlying the findings in their manuscript fully available?

Reviewer #1: Yes

5. Is the manuscript presented in an intelligible fashion and written in standard English?

Reviewer #1: Yes

6. Review Comments to the Author

Reviewer #1: (No Response)

7. PLOS authors have the option to publish the peer review history of their article (what does this mean?). If published, this will include your full peer review and any attached files.

Reviewer #1: No

---

## [Editor Report · Acceptance letter]

14 Jul 2021

PONE-D-21-05218R3 

Mothers’ utilization and associated factors of preconception care in Africa, a systematic review and meta-analysis 

Dear Dr. Tekalign:

I'm pleased to inform you that your manuscript has been deemed suitable for publication in PLOS ONE. Congratulations! Your manuscript is now with our production department. 

Kind regards, 

on behalf of

Dr. Joseph Telfair 

Academic Editor

PLOS ONE